# Biogenic Synthesis of Copper Nanoparticles Using Bacterial Strains Isolated from an Antarctic Consortium Associated to a Psychrophilic Marine Ciliate: Characterization and Potential Application as Antimicrobial Agents

**DOI:** 10.3390/md19050263

**Published:** 2021-05-08

**Authors:** Maria Sindhura John, Joseph Amruthraj Nagoth, Marco Zannotti, Rita Giovannetti, Alessio Mancini, Kesava Priyan Ramasamy, Cristina Miceli, Sandra Pucciarelli

**Affiliations:** 1School of Biosciences and Veterinary Medicine, Biosciences and Biotechnology Division, University of Camerino, 62032 Camerino, Italy; mariasindhura.john@unicam.it (M.S.J.); josephamruthraj.nagoth@unicam.it (J.A.N.); alessio.mancini@unicam.it (A.M.); kesava.ramasamy@unicam.it (K.P.R.); cristina.miceli@unicam.it (C.M.); 2School of Sciences and Technology, Chemistry Division, University of Camerino, 62032 Camerino, Italy; rita.giovannetti@unicam.it

**Keywords:** green synthesis, biomaterials, metal, antibiotics, nanotechnology

## Abstract

In the last decade, metal nanoparticles (NPs) have gained significant interest in the field of biotechnology due to their unique physiochemical properties and potential uses in a wide range of applications. Metal NP synthesis using microorganisms has emerged as an eco-friendly, clean, and viable strategy alternative to chemical and physical approaches. Herein, an original and efficient route for the microbial synthesis of copper NPs using bacterial strains newly isolated from an Antarctic consortium is described. UV-visible spectra of the NPs showed a maximum absorbance in the range of 380–385 nm. Transmission electron microscopy analysis showed that these NPs are all monodispersed, spherical in nature, and well segregated without any agglomeration and with an average size of 30 nm. X-ray powder diffraction showed a polycrystalline nature and face centered cubic lattice and revealed characteristic diffraction peaks indicating the formation of CuONPs. Fourier-transform infrared spectra confirmed the presence of capping proteins on the NP surface that act as stabilizers. All CuONPs manifested antimicrobial activity against various types of Gram-negative; Gram-positive bacteria; and fungi pathogen microorganisms including *Escherichia coli*, *Staphylococcus aureus*, and *Candida albicans*. The cost-effective and eco-friendly biosynthesis of these CuONPs make them particularly attractive in several application from nanotechnology to biomedical science.

## 1. Introduction

With the beginning of the 21st century, nanobiotechnology entered the scientific spot-light as a discipline for innovative materials and applications. Nanoparticles (hereafter called NPs) are becoming the fundamental building blocks of nanotechnology. Their small dimensions and high surface area to volume enable them to exhibit novel chemical and physical properties. Consequently, NPs can be used in applications that are different from those of their bulk materials, including but not limited to electrical resistivity and conductivity, chemical reactivity, and diverse and versatile biological processes [1,2]. Most research work in this field focused on silver and gold NPs. However, special interest has been taken in other metal NPs because these particles are being widely used as industrial catalysts, in chemical sensing devices, in medical applications, in cosmetics, and as microelectronics [1,2,3,4]. Among these, interest is growing for NPs from copper due to the attractive physical and chemical properties. Specifically, there are different types of copper NPs that can be synthesized via multiple methods: CuNPs, cupric oxide nanoparticles (CuONPs), and cuprous oxide nanoparticles (Cu_2_ONPs) [5,6,7,8]. In particular, CuONPs are p-type oxide semiconductors that, for its catalytic, optical, magnetic, and electrical properties, are largely used as catalysts; as sensors; in optoelectronics; as photocatalyst; and in antibacterial, antifungal, antiviral, anticancer, antioxidant, or drug delivery applications, among other [9,10]. In addition, CuONPs are significantly less toxic, with values L50 values reported between 64 and 840 mg/L vs. between 0.22 and 24 mg/L for CuNPs [11]. For the synthesis of CuNPs, many methods have been employed such as thermal evaporation, chemical synthesis, electrochemical synthesis, solvothermal route, and vapor–liquid–solid growth [5,6,7,8,9,10]. However, physical methods have some disadvantages such as the requirement of expensive and complicated vacuum techniques. Chemical synthesis limits NP applications in clinical fields due to the usage of toxic chemicals that may lead to environmental and biological hazards. By contrast, CuNP green synthesis is eco-friendly and cost effective and does not use toxic chemicals, which make CuNPs attractive in biomedical applications. 

Cu is an essential micronutrient for living cells because it is a constituent of many metalloenzymes including cytochrome-c oxidase and superoxide dismutase (SOD) [12]. However, Cu can also generate reactive oxygen species [13,14], it can be poisonous at high concentrations, and can reduce microorganism growth. It is well established that, when microbes are kept in a toxic metal environment, they evolve mechanisms to survive in harsh conditions by transforming toxic metal ions into their corresponding nontoxic forms such as metal sulfide/oxides. For example, a new *Alcanivorax* sp. isolated from a shallow hydrothermal vent was resistant to copper toxicity [15]. A large group of biological resources such as bacteria, yeasts, fungi, algae, and plants can be used for the synthesis of NPs from metal ions; however, the detailed mechanisms involved in nanoscale transformation are not well established. Among all biological systems used until now, bacteria have acquired significant attention as they are easy to culture, are able to produce extracellular NPs with easy downstream processing (as purification steps), and have short generation times for NP synthesis. Furthermore, a large group of biological resources such as bacterial biomolecules acts as a reducing and stabilizing agent for NP synthesis, limiting particle growth, and prohibiting agglomeration, resulting in the formation of desired NPs. In this regard, copper oxide nanoparticles have different applications, such as antibacterial, antifungal, antiviral, anticancer, antioxidant, and drug delivery applications [10]. 

In the present study, we report the biosynthesis of CuONPs at low temperatures from bacterial strains isolated from a consortium associated with the Antarctic ciliate *Euplotes focardii* [16]. This microorganism is a free-swimming ciliate, endemic of the oligotrophic coastal sediments of the Terra Nova Bay [17]. All bacterial strains are identified as *Marinomonas* (MM) [18], *Rhodococcus* (RH), *Pseudomonas* (PM) [19,20,21], *Brevundimonas* (BM), and *Bacillus* (BC). All were named with the ”ef1” suffix (ef stands for *Euplotes focardii* and 1 indicates that is the first strain of the genus isolated from this organism). 

Although Antarctica is regarded as the last uncontaminated continent, it is not completely free from pollution [22]. Despite isolation of the continent by natural barriers such as circumpolar atmospheric and oceanic currents [23], contaminants such as heavy metals, pesticides, and other persistent organic pollutants (POPs) could reach Antarctica via long range atmospheric transport (LRAT) from other continents in the southern hemisphere and even beyond [24]. Previous work reported evidence that bacteria from Antarctica developed resistance to heavy metals [25,26]. Our results support this evidence since these bacterial strains shows resistance to up to 5 mM of CuSO_4_ and produced CuONPs. Our study highlights an efficient strategy in obtaining bionanomaterials that can be used as antibiotics against a large number of drug-resistant pathogens bacteria, which has created serious concern across the globe due to the limited choices in antibiotic treatment [27].

## 2. Results and Discussion

### 2.1. Copper Tolerance and Growth Assessment for the Bacterial Strains

As a first step in this work, we assessed the Cu tolerance of all bacterial strains under study. Table 1 and Table 2 report the bacterial growth assessments at different CuSO_4_ concentrations and the maximum tolerated concentrations (MTCs) of heavy metals, respectively. Our results indicate that all strains tolerate CuSO_4_ up to 3.5–4 mM.

We also monitored bacterial growth in the presence of increasing CuSO_4_ concentrations for each strain (Appendix A): increasing copper concentrations decreased the growth rate of all bacteria tested. The highest growth inhibition effect is visible at concentrations above 3.5 mM, in particular, for Pseudomonas ef1 and Bacillus ef1 (Appendix A). 

### 2.2. Biosynthesis of CuNPs

All of the tested bacteria showed resistance to Cu up to 3.5 mM; thus, we reasoned that these strains may be able to synthesize copper NPs. With the addition of 1 mM CuSO_4_ (final concentration) in the reaction medium, we observed a gradual change in the solution color from cyan to brown over a 48 h period of time (Appendix A). A similar change in color has been reported after the addition of 5 mM CuSO_4_ to a flask containing *Morganella* sp. [28], or three different species of *Penicillium* and the white-rot fungus *Stereum hirsutum* [29]. Therefore, our result suggests the formation of copper NPs from CuSO_4_ through microbial metabolisms.

### 2.3. Ultraviolet–Visible Absorption Spectroscopy (UV–Vis), Dynamic Light Scattering (DLS) Analysis, and Zeta Potential Measurements of CuNPs

UV–vis spectral analysis is the most important analysis method to detect the surface plasmon resonance (SPR) property of biosynthesized CuNPs. We applied UV–vis spectroscopy to all of the samples obtained from the different strains: a sharp peak with maximum absorption in the range of 381–383 nm was recorded in each sample and can be attributed to the formation of CuNPs [30] (Appendix A). 

Our results are in agreement with previous reports on bacterial synthesized CuNPs. Indeed, broad absorption spectra peaks were observed at around 365 nm for CuNPs synthesized from *Escherichia coli* [31], at 310 nm for *Eichhornia crassipes* [32], and at 360 nm for CuNPs synthesized from Ixora coccinea leaf extracts [30]. 

Dynamic light scattering (DLS) was also performed to determine the size distribution and zeta potential for biosynthesized CuNPs. All of the average diameters are reported in Figure 1 for all of the different bacterial strains.

Similarly, the zeta potential values of CuNPs are in the range from −23.2 mV to −33.8 mV (Table 3). The zeta potential data demonstrated that biosynthesized NPs were stable in a liquid medium, and therefore, the tested bacterial strains represent a good source for production of narrow-sized CuNPs. The stability may be due to NP capping by biomolecules produced as byproducts. The higher negative zeta potential values obtained, thus, confirmed the repulsion between particles to thwart agglomeration.

### 2.4. XRD and FTIR Analyses

The powered XRD analysis of synthesized CuNPs with different strains was applied in order to investigate the crystalline phase of Cu nanostructures. In Figure 2, all XRD profiles of CuNPs are reported in comparison with simulation data (black line defined as theoretical). The obtained spectra reveal characteristic diffraction peaks indicating the formation of CuO in monoclinic and crystalline phase (the specific explanations are reported in Appendix B). In the CuO monoclinic structure, each Cu atom is situated at the center of four oxygen atoms positioned at the vertices of a rectangle with oxygen atoms at the center of a tetrahedron of Cu atoms [31]. The slight differences in peak positions with respect to the simulation data indicate that the different coatings of the obtained nanoparticles promote defects in the crystals, giving different crystalline orientations with respect to the theoretical ones. In addition, the XRD pattern demonstrate the absence of impurity and of peaks related to Cu(OH)_2_ or Cu_2_O phases (see Appendix B for more details).

FTIR spectroscopy was carried out to verify the possible involvement of functional groups or biomolecules in CuONP formation and stabilization. This technique is a powerful tool used to identify the chemical bonds in a molecule by producing an IR spectrum that is similar to a molecular fingerprint. The FTIR spectra of CuONPs biosynthesized by MM-ef1, RH-ef1, PS-ef1, BM-ef1, and BC-ef1 were obtained in the range between 400 and 4000 cm^−1^ (Appendix A). 

The IR spectra suggested that the biomolecules interacted with the biosynthesized CuONPs. The distinct bands analyzed indicated the presence of –OH, –NH, and –CH_2_ scissor vibrations of aliphatic compounds and C=C bonds inside the biomolecules. From the analysis of the peaks, a carbonyl group (C=O) of the amide functional group was also detected. Therefore, the presence of carbonyl and NH groups is important for stabilization of the nanoparticles. In fact, the doublet of electrons present on both groups can be useful for the electrostatic stabilization of CuONP nanoparticles and thus to function as a capping agent [32].

### 2.5. Transmission Electron Microscopy (TEM)

Controlling NP size distribution is important for many applications, in particular for antimicrobial activity. We applied TEM analysis (Figure 3) to obtain additional insights in the morphology and size of the CuONPs from all of the strains. 

TEM micrographs of all bacterial CuONPs showed that the particles are mono-dispersed, spherical/ovoidal shapes. The micrographs suggested particle sizes around 10 nm to 70 nm, with the average size being 40 nm. The irregular shape and change in dimensions observed using different bacteria during the synthesis depends by the different bacterial metabolisms and of the secondary compounds produced that surround the nanoparticles. These compounds act as capping and dispersing agents influencing the shape and dimension of the nanostructures. All of the nanoparticles are well segregated except for the nanoparticles produced by *Marinomonas* ef1, where larger aggregates are observed (Figure 3). 

### 2.6. Antimicrobial Activity of the Bacterial CuONPs

It has been reported that CuNPs of about 30 nm are more active against pathogens than those of smaller size [33]. CuONPs have received much attention in a wide range of applications, including their use as antimicrobial agents [9,34,35,36,37]. 

To test the potential antimicrobial activity of our biosynthesized CuONPs, we performed the disk diffusion test against *Staphylococcus aureus*, *Escherichia coli* (EC), *Klebsiella pneumoniae* (KP), *Pseudomonas aeruginosa* (PA), *Proteus mirabilis* (PrM), *Citrobacter koseri* (CK), *Acinetobacter baumanii* (AB), *Serratia marcescens* (SM), *Candida albicans* (CA), and *Candida parapsilosis* (CP). 

The results of the disk diffusion test are reported in Figure 4, in which the presence of a clear zones around the CuONP disk is clearly visible, suggesting that all nanoparticles were able to inhibit the growth of bacterial and fungal pathogens.

All of the results for inhibition zone variation are summarized in Table 4 as a comparison with negative controls. 

We further evaluated the CuONP antibacterial activity by estimating the minimum inhibitory concentration (MIC) and minimum bactericidal concentration (MBC) values using the plate microdilution method [38]. The MIC is defined as the lowest concentration of antibacterial agent needed that inhibits the growth of bacteria. As showed in Table 5 and Appendix A, the CuONPs synthesized by all bacteria showed an MIC against the Gram-negative bacteria from 3.12 to 25 μg/mL, against Gram-positive bacteria from 12.5 to 25 μg/mL, and against fungi from 12.5 to 25 μg/mL. MBC is defined as the lowest concentration of an antibacterial agent that kills the bacteria (no growth was observed on the agar plate). In the study, MBC values ranged from 12.5 to 25 μg/mL. These analyses showed that these CuONPs possess strong antimicrobial activity. The different MIC and MBC values could be attributed to the different shapes and dimensions of the nanoparticles determining their antibacterial activity.

It was observed that the bactericidal property of CuONPs is due to the generation of a large number of small molecules containing reactive oxygen species (ROS) by the nanoparticles attached to the bacterial cells. Although cells can detoxify ROS species within certain limits by promoting antioxidant activity such as specific enzymes or small molecules (e.g., ascorbic acid), the formation and accumulation of ROS species largely increased when the cell was constitutively exposed to intracellular oxidative stress [36]. 

## 3. Materials and Methods

### 3.1. Culture and Chemicals

All bacterial strains used in this work were isolated from a consortium associated with the Antarctic ciliate *E. focardii* and identified as *Marinomonas*, *Rhodococcus, Pseudomonas*, *Brevundimonas*, and *Bacillus* [17,18,20]. All strains were grown at 22 °C on agarized or liquid Luria–Bertani (LB) medium (tryptone 10g/L, yeast extract 5 g/L, and NaCl 5 g/L). All media were purchased from Liofilchem. Analytical grade copper (II) sulfate pentahydrate salt, CuSO_4_·5H_2_O, and the other chemicals were purchased from Sigma Aldrich (Sigma Aldrich, St. Louis, MO, USA).

### 3.2. Determination of Copper Maximum Tolerated Concentrations (MTCs)

Copper (Cu) maximum tolerated concentrations (MTCs) were determined for each isolate on agarized LB medium in the presence of increasing concentrations (from 0 to 6 mM) of CuSO_4_. With a sterile inoculating loop, each bacterial isolate was streaked on the Cu incorporated agarized LB medium. The plates were incubated at 22 °C and inspected at intervals up to 72 h. The MTCs were noted when the isolate failed to show growth on the plates after three days of incubation. All experimental setups were prepared in triplicate.

### 3.3. Estimation of Bacterial Growth Inhibition by Copper 

Bacterial growth Cu inhibition was determined by monitoring the optical density at regular time intervals at 600 nm using a Shimadzu UV 1800 spectrophotometer. In total, 0.5 mL of an overnight active culture adjusted to OD600 = 0.1 was incorporated into 50 mL of LB medium supplemented with increasing concentrations of CuSO_4_ from 0 up to the maximum concentration tolerated by each bacterium. Inoculated flasks were incubated on a rotary shaker at 22 °C. All experimental setups were prepared in triplicate. 

### 3.4. Biosynthesis of CuNPs

Each strain was inoculated in LB medium (100 mL) and incubated at 22 °C on a rotatory shaker (200 rpm). After 24 h, CuSO_4_·5H_2_O was added to the microbial cell culture to a final concentration of 1mM. The reaction mixture was incubated for 24 to 48 h on a rotatory shaker at 150 rpm at 22 °C. LB medium with 1 mM CuSO_4_ without the organism or heat-killed bacterial cultures were maintained as a control. During the incubation period, the solution color change from blue to dark green (which is indicative of the reduction of CuSO_4_ to CuNPs) was monitored. The formation of CuNPs was also recorded by absorption spectra in the wavelength range of 200–800 nm at room temperature (23 °C) using a Shimadzu UV 1800 spectrophotometer. 

### 3.5. Purification of CuNPs

After incubation, the culture was centrifuged at 5000 rpm at 4 °C for 20 min in a Beckman J2-21 with swinging rotor to separate the cell pellet from the cell-free supernatant. Nanoparticles were purified from both the supernatant and pellet. To recover CuNPs present in the cell-free supernatant, the solution was centrifuged at 17,000× *g* for 15 min in the same centrifuge with a fixed rotor. The CuNPs containing the pellet was then resuspended in double-distilled water (ddH_2_O) and washed twice by repeated centrifugation steps. 

To recover CuNPs from the cell pellet, ultrasonic wave shocks of short durations (15 s) were given to the ddH_2_O-suspended pellet to rupture the microbial cell wall. After sonication, the sample was centrifuged 5000 rpm for 20 min (Beckman J2–21, Fullerton, California) and the NPs were recovered from the supernatant. This step was repeated three times to completely remove the cell debris from the supernatant. To recover CuNPs present in the cell-free supernatant, the solution was centrifuged at 17,000× *g* for 15 min in the same centrifuge with a fixed rotor. After being washed twice with deionized water and dried at 80 °C in an oven, the CuNPs were used for further characterization and experiments.

### 3.6. Dynamic Light Scattering, Zeta Potential Measurement, Transmission Electron Microscopy (TEM), X-ray Diffraction Analysis (XRD), and Fourier-Transform Infrared Spectroscopy (FTIR) Analyses

Zetasizer Nano ZS, (Malvern Instruments Ltd., Malvern, UK) was used to determine the size distribution of particles by measuring dynamic fluctuations of light scattering intensity caused by the Brownian motion of the particles. All measurements were carried out in triplicate with a temperature equilibration time of 2 min at 25 °C. Additionally, NP surface charge was measured using the zeta potential. The morphologies of the biosynthesized CuNPs were observed on a JEOL transmission electron microscope (TEM) system operating at 200 kV (JEM-2100, Hitachi Limited, Tokyo, Japan), with the acquirement of the particle size distribution ascertained from TEM micrographs based on professional software (Nano Measurer 1.2.5). The crystal structure of biosynthesized CuNPs was analyzed by powder X-ray diffraction (XRD) measurements performed using a Rigaku-D/MAX-PC 2500 X-ray diffractometer (Wilmington, MA, USA) with a Cu Kα (λ = 1.5405 Å) in the 2θ range from 20 to 80° at a scan rate of 0.03° S−1. FTIR spectra (Kyoto, Japan) were recorded (Shimadzu IR Affinity-1) to identify the possible interactions between CuNPs and the biomolecule. Analysis was carried out in the range of 400–4000 cm^−1^ at the resolution of 4 cm^−1^.

### 3.7. Kirby–Bauer Disk Diffusion Susceptibility Test, Minimum Inhibitory Concentration (MIC), and Minimum Bactericidal Concentration (MBC) Evaluation

The antibacterial activity of biosynthesized CuNPs was tested against Staphylococcus aureus, Escherichia coli, Klebsiella pneumoniae, Pseudomonas aeruginosa, Proteus mirabilis, Citrobacter koseri, Acinetobacter baumanii, Serratia marcescens, Candida albicans, and Candida parapsilosis. All the strains were cultured in Mueller Hinton broth (MHB) (Merck, Darmstadt, Germany) at 37 °C. The antibacterial activity of CuNPs against the selected bacterial strains was assessed using the Kirby–Bauer disk diffusion susceptibility test method. Using a sterile cotton swab, the bacteria strains were spread on the Mueller–Hinton agar (MHA). The disks were loaded with 25 μL (25 μg) of 1 mg/mL CuNP solution and CuSO_4_ solution (1 mM) and dried. The disks were then placed on the agar plate and incubated at 37 °C. The inhibition zone was observed after 24 h of incubation.

The MIC and MBC estimations of the CuNPs were performed using the method described in the guideline of CLSI 2012 [38]. The MIC test was performed on a 96-well round bottom microtiter plate using standard broth microdilution methods, while the MBC test was performed on MHA plates. The bacterial inoculums were adjusted to the concentration of 0.5 McFarland units. For the MIC test, CuNP stock solution was prepared by ultrasonication in sterilized deionized water to reach 200 μg/mL. A volume of 100 μL of stock solution was serially diluted twofold in 100 μL of MHB in the first row, and finally 100 μL was discarded such that the first well in the row of the microtiter plate contained the highest concentration of CuNPs while the last well of the row contained the lowest concentration. Similarly, CuNPs were prepared in all of the rows. The positive control contained the medium and bacterial inoculums (K^+^), and the negative control contained only the medium (K^−^). The microtiter plate was then incubated at 37 °C for 24 h. The MIC value was defined as the lowest concentration of antibacterial agents that inhibits the growth of bacteria. The MBC was taken as the lowest concentration of antibacterial agents that completely kills the bacteria. To check MBC, the suspension from each well of the microtiter plates was plated onto the MHA plate and were incubated at 37 °C for 24 h. The MBC value was taken as the lowest concentration with no visible growth on the MHA plate.

## 4. Conclusions

In this study, we reported the production of monodisperse, small, and highly pure bio-CuONPs using the green reduction of CuSO_4_ at low temperatures, such as 22 °C, by using five Antarctic bacterial strains. The ability of these bacteria to synthesize CuONPs may represent a defense mechanism against this heavy metal. The results support the evidence that these bacterial strains are resistant to up to 5 mM of CuSO_4_. All of the nanoparticles were fully characterized and tested for their antibacterial activity. 

This study confirms that the tested Antarctic bacterial strains can be exploited in bioremediation to remove copper contamination from the environment and in the production of antibiotics against various types of pathogenic Gram-negative and Gram-positive bacteria, and fungi including *Escherichia coli*, *Staphylococcus aureus*, and *Candida albicans*. These results showed that the cost-effective and eco-friendly biosynthesis of these CuONPs make them particularly attractive in several applications including biomedical science.

## 5. Patents

The results of this paper are related to patent numbers 102019000014121 and 102019000024493. *Marinomonas* sp. ef1 and *Rhodococcus* sp. ef1 have been deposited at the Istituto Zooprofilattico Sperimentale della Lombardia e dell’Emilia Romagna “Bruno Ubertini”—IZSLER according to the Budapest treaty under access No. DPS RE RSCIC 17 and DPS RE RSCIC 4. *Brevundimonas* sp ef1 and *Bacilus* sp ef1 have be deposited under access No. DPS RE RSCIC 23 and DPS RE RSCIC 24.

## Figures and Tables

**Figure 1 marinedrugs-19-00263-f001:**
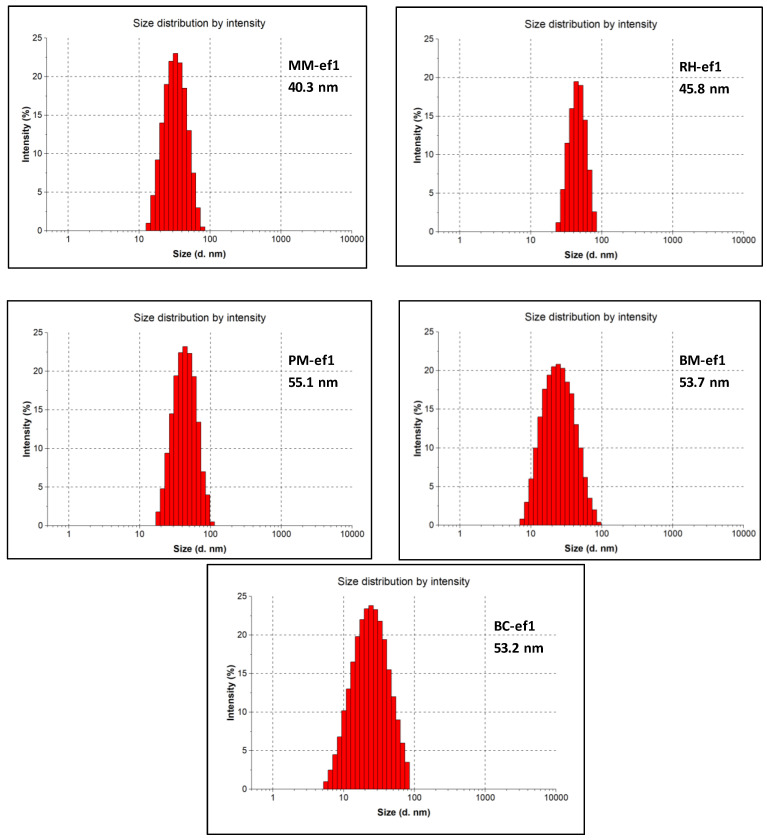
DLS of bio-CuNPs synthesized from the different bacterial strains: *Marinomonas* (MM), *Rhodococcus* (RH), *Pseudomonas* (PM), *Brevundimonas* (BM), and *Bacillus* (BC).

**Figure 2 marinedrugs-19-00263-f002:**
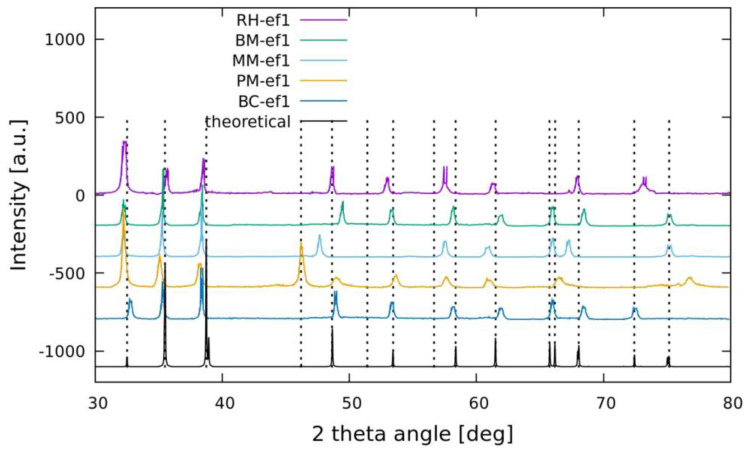
XRD profile of biosynthesized CuONPs. The obtained spectra reveal characteristic diffraction peaks indicating the formation of CuO in all bacterial NPs in comparison with simulation peaks (black line defined as the theoretical peaks).

**Figure 3 marinedrugs-19-00263-f003:**
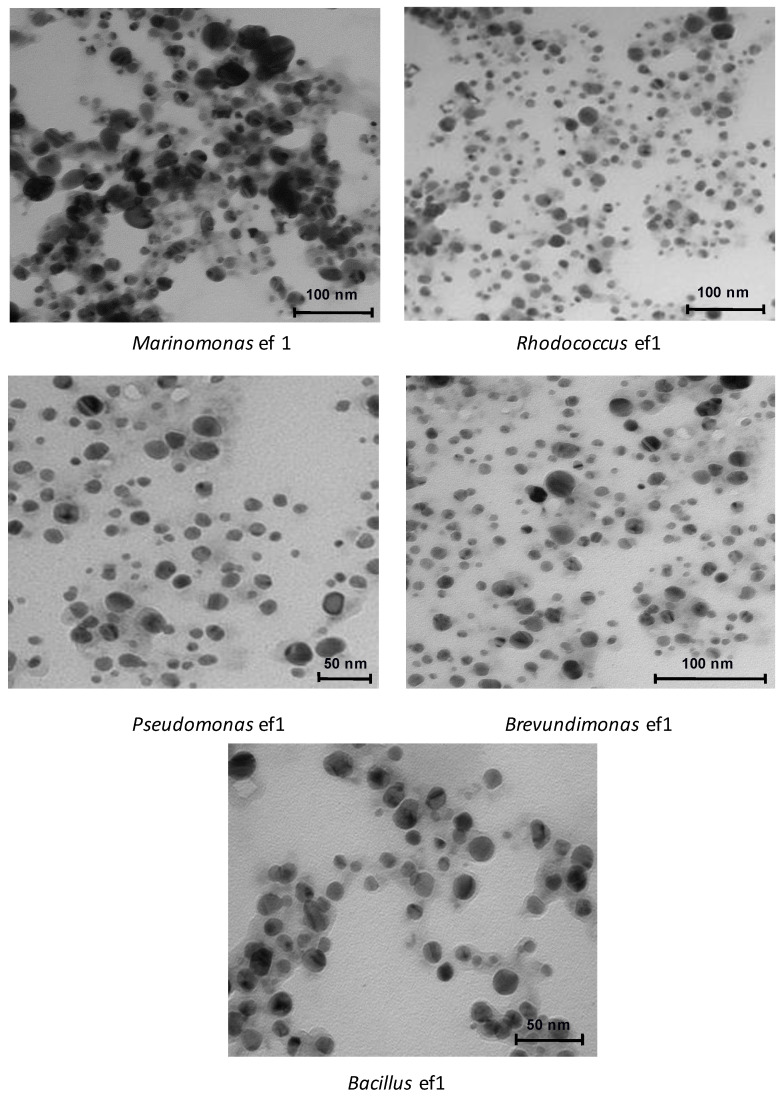
TEM images of biosynthesized CuONPs from the five bacterial strains: *Marinomonas* ef1, *Rhodococcus* ef1, *Pseudomonas* ef1, *Brevundimonas* ef1, and *Bacillus* ef1.

**Figure 4 marinedrugs-19-00263-f004:**
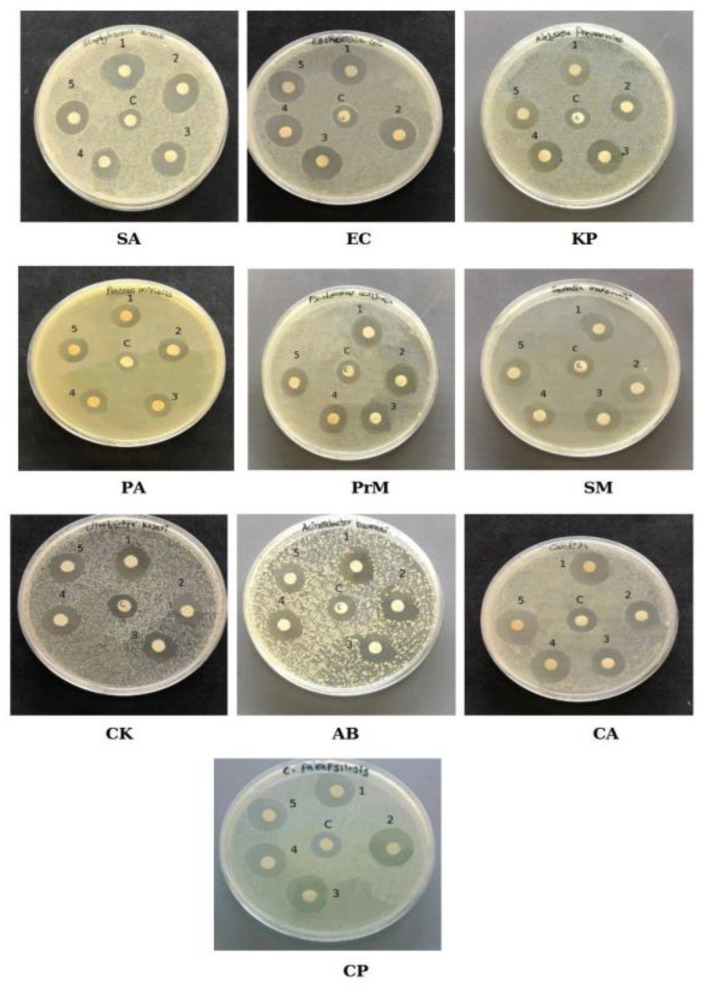
Antibacterial activity of bio-CuONPs against pathogenic bacteria: Staphylococcus aureus (SA), Escherichia coli (EC), Klebsiella pneumoniae (KP), Pseudomonas aeruginosa (PA), Proteus mirabilis (PrM), Serratia marcescens (SM), Citrobacter koseri (CK), Acinetobacter baumanii (AB), Candida albicans (CA), and Candida parapsilosis (CP). Kirby–Bauer disk diffusion test. 1. MM-ef1, 2. RH-ef1, 3. BM-ef1, 4. PM-ef1, 5. BC-ef1, and C—control (1 mM of CuSO_4_).

**Table 1 marinedrugs-19-00263-t001:** Growth assessment of bacteria with various concentrations of CuSO_4_. High growth: +++, medium growth: ++, low growth: +, and no growth: −.

	CuSO_4_ Concentration (mM)												
Organisms	0.0	0.5	1	1.5	2	2.5	3	3.5	4	4.5	5	5.5	6
*Marinomonas* ef1	+++	+++	+++	+++	+++	+++	++	++	++	++	+	−	−
*Rhodococcus* ef1	+++	+++	+++	+++	+++	+++	+++	++	+	+	−	−	−
*Pseudomonas* ef1	+++	+++	+++	+++	++	++	+	+	−	−	−	−	−
*Brevundimonas* ef1	+++	+++	+++	+++	+++	++	++	++	++	+	+	−	−
*Bacillus* ef1	+++	+++	+++	+++	+++	++	++	+	+	−	−	−	−

**Table 2 marinedrugs-19-00263-t002:** Maximum tolerated CuSO_4_ concentrations (MTCs).

Organisms	CuSO_4_ (mM)
*Marinomonas* ef1	5
*Rhodococcus * ef1	4.5
*Pseudomonas* ef1	3.5
*Brevundimonas* ef1	5
*Bacillus* ef1	4

**Table 3 marinedrugs-19-00263-t003:** Zeta potential of CuNPs obtained from different strains.

Organisms	Zeta Potential (mV)
*Marinomonas* ef1	–23.2
*Rhodococcus* ef1	–33.8
*Pseudomonas* ef1	–33.1
*Brevundimonas* ef1	–33.6
*Bacillus* ef1	–25.1

**Table 4 marinedrugs-19-00263-t004:** Inhibition zone variation due to antibacterial activity of bio-CuONPs.

	MM-ef1	RH-ef1	BM-ef1	PM-ef1	BC-ef1	CuSO_4_
	**Gram-Positive Bacteria**					
*SA*	16 ± 0.2 *	16 ± 0.3 *	16 ± 0.1 *	13 ± 0.4 *	15 ± 0.2 *	9 ± 0.3 °
	**Gram-Negative Bacteria**					
*EC*	15 ± 0.2 *	16 ± 0.4 *	17 ± 0.2 *	16 ± 0.3 *	16 ± 0.2 *	10 ± 0.1 °
*KP*	16 ± 0.1 *	15 ± 0.1 *	17 ± 0.3 *	16 ± 0.4 *	16 ± 0.3 *	10 ± 0.3 °
*PM*	16 ± 0.3 *	17 ± 0.4 *	16 ± 0.4 *	15 ± 0.4 *	15 ± 0.2 *	11 ± 0.2 °
*PrM*	15 ± 0.1 *	15 ± 0.2 *	14 ± 0.2 *	15 ± 0.2 *	15 ± 0.4 *	10 ± 0.2 °
*CK*	16 ± 0.3 *	15 ± 0.3 *	15 ± 0.2 *	16 ± 0.2 *	16 ± 0.3 *	10 ± 0.1 °
*AB*	16 ± 0.2 *	17 ± 0.4 *	15 ± 0.1 *	16 ± 0.2 *	15 ± 0.2 *	11 ± 0.2 °
*SM*	15 ± 0.2 *	15 ± 0.2 *	15 ± 0.2 *	14 ± 01 *	14 ± 0.2 *	10 ± 0.3 °
	**Fungi**					
*CA*	18 ± 0.4 *	16 ± 0.4 *	16 ± 0.4 *	17 ± 0.4 *	18 ± 0.4 *	11 ± 0.2 °
*CP*	17 ± 0.4 *	19 ± 0.2 *	17 ± 0.4 *	16 ± 0.4 *	16 ± 0.4 *	11 ± 0.2 °

Inhibition zone (Ø, in mm) with * CuONPs and with ° CuSO_4_ (negative control).

**Table 5 marinedrugs-19-00263-t005:** MIC and MBC values (μg/mL) of the CuONPs synthesized by MM-ef1, RH-ef1, PM-228 ef1, BM-ef1, and BC-ef1 against pathogenic bacteria.

	MM-ef1		RH-ef1		BM-ef1		PM-ef1		BC-ef1	
	MIC	MBC	MIC	MBC	MIC	MBC	MIC	MBC	MIC	MBC
	**Gram Positive Bacteria**									
*SA*	12.5 ± 0.2	25 ± 0.4	12.5 ± 0.2	25 ± 0.2	25 ± 0.2	25 ± 0.4	12.5 ± 0.4	25 ± 0.2	12.5 ± 0.4	12.5 ± 0.1
	**Gram Negative Bacteria**									
*EC*	12.5 ± 0.2	25 ± 0.3	12.5 ± 0.4	25 ± 0.3	25 ± 0.4	25 ± 0.3	25 ± 0.3	25 ± 0.2	12.5 ± 0.2	12.5 ± 0.2
*KP*	12.5 ± 0.1	25 ± 0.5	6.25 ± 0.3	12.5 ± 0.2	12.5 ± 0.2	25 ± 0.4	12.5 ± 0.2	25 ± 0.4	6.25 ± 0.2	6.25 ± 0.2
*PM-sp*	6.25 ± 0.2	12.5 ± 0.2	12.5 ± 0.2	25 ± 0.4	12.5 ± 0.2	25 ± 0.3	12.5 ± 0.1	12.5 ± 0.4	12.5 ± 0.3	12.5 ± 0.4
*PM*	3.12 ± 0.1	6.25 ± 0.2	6.25 ± 0.2	12.5 ± 0.1	6.25 ± 0.1	12.5 ± 0.2	6.25 ± 0.2	12.5 ± 0.2	6.25 ± 0.1	12.5 ± 0.1
*CK*	6.25 ± 0.2	6.25 ± 0.1	12.5 ± 0.3	12.5 ± 0.2	12.5 ± 0.2	25 ± 0.4	6.25 ± 0.1	12.5 ± 0.4	12.5 ± 0.4	12.5 ± 0.1
*AB*	12.5 ± 0.2	12.5 ± 0.2	12.5 ± 0.2	12.5 ± 0.2	12.5 ± 0.2	25 ± 0.4	12.5 ± 0.2	12.5 ± 0.2	12.5 ± 0.2	12.5 ± 0.4
*SM*	6.25 ± 0.1	12.5 ± 0.4	3.12 ± 0.1	12.5 ± 0.2	6.25 ± 0.1	12.5 ± 0.2	6.25 ± 0.1	12.5 ± 0.2	6.25 ± 0.2	12.5 ± 0.2
	**Fungi**									
*CA*	25 ± 0.4	25 ± 0.5	12.5 ± 0.2	25 ± 0.4	12.5 ± 0.2	25 ± 0.4	25 ± 0.2	25 ± 0.4	12.5 ± 0.2	25 ± 0.3
*CP*	12.5 ± 0.4	25 ± 0.3	25 ± 0.2	25 ± 0.2	12.5 ± 0.1	25 ± 0.2	12.5 ± 0.3	25 ± 0.1	6.25 ± 0.2	12.5 ± 0.2

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
