# Peer review of "Biogenic Synthesis of Copper Nanoparticles Using Bacterial Strains Isolated from an Antarctic Consortium Associated to a Psychrophilic Marine Ciliate: Characterization and Potential Application as Antimicrobial Agents"

_marinedrugs, 2021, doi:10.3390/md19050263_

Round 1
Reviewer 1 Report
Very interesting topic but there are a lot of mistakes in the work that need to be corrected. Bacteria are misspelled and should be written in italic throughout the text. an abbreviation of the Latin names of bacteria should also be commonly written.
FIG. 3. the pictures should be enlarged and marked differently and the text with explanations should be given below the picture. Did you use control in the experiments? nanoparticles obtained by a different process? Nanoparticles look very uneven, can you explain that? Is there a difference between nanoparticles formed due to the action of different bacterial cells and why?
I don't know if it is necessary to list all the pictures and also the insufficient quality and small format. If you want to display the image, enlarge and highlight them. I think they are superfluous, but if the authors evaluate differently, let them change the way of presentation and marking. And below this picture you need to write the bacteria correctly.
Did you have variations in MIC and MBC values?
Is 15 seconds in an ultrasonic bath enough to break up cells? Did you control it and how? was there a difference between nanoparticles isolated from the supernatant and bacterial cell pellets?
Author Response
We thank the Reviewer for the helpful suggestions.
Comment: Very interesting topic but there are a lot of mistakes in the work that need to be corrected. Bacteria are misspelled and should be written in italic throughout the text. an abbreviation of the Latin names of bacteria should also be commonly written.
Response: In the revised version of the paper, we corrected all the bacterial names.
Comment: FIG. 3. the pictures should be enlarged and marked differently and the text with explanations should be given below the picture.
Response: in the revised version of the paper we provided an enlarged version of fig. 3 with more detailed explanation in the figure legend
Comment: Did you use control in the experiments? nanoparticles obtained by a different process?
Response: Yes we did. Indeed, nanoparticles are obtained by single process that is cell mediated (intracellular) synthesis. LB medium with 1mM CuSO4 without the microorganism and heat killed bacterial cultures were maintained as a control. These controls are used to prove that the bacterial cells and their extracts mediated the NP synthesis and confirm that the medium components have no role in the reduction of Cu precursor and formation of nanoparticles. The use of controls was explained in the first version of the manuscript in chapter 3.4. Biosynthesis of CuNPs
Comment: Nanoparticles look very uneven, can you explain that? Is there a difference between nanoparticles formed due to the action of different bacterial cells and why?
Response: We thank the reviewer for the question.
The nanoparticles have all spherical/ovoidal shapes as reported in the TEM images of Figure 3. The irregular shape and change in dimensions observed by using different bacteria during the synthesis, depends by the different bacterial metabolism and of the produced secondary compounds that surround the nanoparticles. These compounds act as capping and dispersing agents influencing the shape and dimension of the nanostructures. In the manuscript the explanation about shapes and different dimension have been added in the section 2.5.
Comment: I don't know if it is necessary to list all the pictures and also the insufficient quality and small format. If you want to display the image, enlarge and highlight them. I think they are superfluous, but if the authors evaluate differently, let them change the way of presentation and marking. And below this picture you need to write the bacteria correctly.
Response: We prefer to show all TEM pictures. In the revised version of the paper the Figure 3 has been modified according to the reviewer suggestion and the name of bacteria has been added.
Comment:Did you have variations in MIC and MBC values?
Response: Yes, we have variations. This might be due to the presence in the sample of polydisperse nanoparticles with different sizes. Size and shape of the nanoparticles determine the antibacterial activity. This explanation has been added on the section 2.6.
Comment: Is 15 seconds in an ultrasonic bath enough to break up cells? Did you control it and how?
Response: Ultrasonic wave shocks of short duration (15 s) were given to rupture microbial cell wall. High power ultrasound at frequencies around 20kHz is capable of killing bacteria and for many years has been used as standard technique in microbiology for the disruption of living cells to release their contents.
The control we performed was to check if dark nanoparticles were realeased in the supernatant, demonstrating that bacterial cells walls were broken.
Comment: was there a difference between nanoparticles isolated from the supernatant and bacterial cell pellets?
Response: We did not observe any difference between supernatant and bacterial cell pellets nanoparticles. We believe that supernatant NPs are the ones synthesized intracellularly and then released outside the cell once they formed and accumulated in the cytoplasm.
Reviewer 2 Report
The manuscript „ Biogenic synthesis of copper nanoparticles using bacterial strains isolated from an Antarctic consortium associated to a psychrophilic marine ciliate: characterization and potential application as antimicrobial agents” evaluates the capacity of Antarctic bacterial strains to synthesize CuONPs and tests the antimicrobial activity of biosynthesized nanoparticles. As several biological sources have been used to produce nanoparticles, authors should indicate the potential advantage of the Antarctic bacterial consortium.
The CuONPs were purified from the supernatant of the cell cultures and by rupturing the cells. Can any differences be detected between the cell-derived and the “free” nanoparticles?Is there any evidence that the antimicrobial activity of CuONPs is due to ROS production in this case?Was the activity of the cell-free extract of the NP-producing bacterial strains tested against bacteria and fungi?
Some comments:
1. All the genus and species name have to be written in italic style.
2. Page 2, Line 58: My suggestion is to change “nutrient” for “micronutrient”
3. What was the unit measuring the inhibition zones?
4. The digits should be subscript characters in the chemical formulas.
The studies conducted well, however spacing and spelling errors have been made in the manuscript that the authors need to rectify.
Author Response
We thank the Reviewer for the precious suggestions.
The manuscript “Biogenic synthesis of copper nanoparticles using bacterial strains isolated from an Antarctic consortium associated to a psychrophilic marine ciliate: characterization and potential application as antimicrobial agents” evaluates the capacity of Antarctic bacterial strains to synthesize CuONPs and tests the antimicrobial activity of biosynthesized nanoparticles.
Comment: As several biological sources have been used to produce nanoparticles, authors should indicate the potential advantage of the Antarctic bacterial consortium.
Response: The advantages of using Antarctic bacteria for nanoparticles synthesis is the efficient synthesis at low temperatures. These new strains are able to synthesize copper nanoparticles at 22 °C, whereas mesophilic bacteria reported to be able to synthesize nanoparticles need temperature up to 37 °C (https://doi.org/10.1016/j.procbio.2016.08.008).
In the Introduction and in the conclusion of the new version of the paper the low temperature synthesis was emphasized.
Comment: The CuONPs were purified from the supernatant of the cell cultures and by rupturing the cells. Can any differences be detected between the cell-derived and the “free” nanoparticles?
Response: The presence of NPs in the supernatant is probably due to the efflux of nanoparticles from the bacterial cell to the outside. We collected nanoparticles from supernatant after cell rupture by sonication. We did not note any difference between cell-derived and “free” nanoparticles, since both derive from intracellular synthesis and modifications during excretion are not reported.
Comment: Is there any evidence that the antimicrobial activity of CuONPs is due to ROS production in this case?
Response: Even though we do not have direct evidence that CuONPs antimicrobial activity is due to ROS production, this correlation has been reported in 10.1002/smll.201200772 (reference 36 in the manuscript)
Comment: Was the activity of the cell-free extract of the NP-producing bacterial strains tested against bacteria and fungi?
Response: Yes, we tested cell-free extract of the NP-producing bacterial strains against bacteria and fungi. Only Bacillus ef1 produced a small inhibition zone on Staphylococcus aureus. This result is not reported in the paper.
Some comments:
1. All the genus and species name have to be written in italic style.
2. Page 2, Line 58: My suggestion is to change “nutrient” for “micronutrient”
Response: We thank the reviewer for the indications, in the revised version we followed these suggestions.
3. What was the unit measuring the inhibition zones?
Response: the zone of inhibition is measured in terms of millimeters (mm). This information has been added in the revised version of the paper.
4. The digits should be subscript characters in the chemical formulas.
Response: we corrected as suggested.
The studies conducted well, however spacing and spelling errors have been made in the manuscript that the authors need to rectify.
Response: we corrected as suggested.
Reviewer 3 Report
The submitted paper describes biogenic synthesis of copper nanoparticles using bacterial strains isolated from an Antarctic consortium associated to a psychrophilic marine ciliate Euplotes focardii.
The manuscript appears well-written and full of details, that allow the understanding of each step, even to non-experts. All experiments were very wide planed and fulfil expectation for these type of paper.
To sum up, I recommend this paper for publication after minor improvements listed below.
I also have two questions for the Authors:
- Out of curiosity, I wonder if the authors have tested these bacterial strains produced CuONPs for cytotoxicity? Any data showing low toxicity especially on human cells would be nice to include and strengthen their potential as future antibacterial agents. The authors only gave in their conclusions link to the publication where L50 values for C. reinhardtii, E. coli, D. magna and embryonic zebrafish were estimated.
- As the authors conclude that antimicrobial activity is likely due to ROS induction, a simple test with DHE or H2DCFDA would be a good complement to the microbiological analysis. Authors should think about this in the future.
Suggested changes
- comparing Table 1 with Table 2, it seems to me that the MTCs for Rhodococcus ef1 should be 4.5 mM and for Brevundimonas ef1 - 5 mM
- all bacterial species should be in italics (line 97, 103, 104, 116, 201-204, 304-306, Fig. 4)
- Fig. 1, the X and Y axis signatures are not very legible and should be larger and on the graph for BC-ef1 s “nm" is missing
- line 219, I would remove "by serial dilution"
- authors should decide whether to use μg/ml or μg/mL (line 224 and 228)
- Table 5, the description is illogical, should be “MIC and MBC values (μg/ml) of the CuONPs synthesized by MM-ef1, RH-ef1, PM-228 ef1, BM-ef1 and BC-ef1 against pathogenic bacteria“
- Methods - MIC evaluation, it is more logical that positive control is medium with inoculum and negative control only medium. It should be changed
- Appendix A, Table “IR assignments signatures” - should not be in italics
- Supplementary Figures:
Fig. S1 Not for all strains the CuSO4 concentration was 0.5-4 mM. The concentrations are given in the diagram, therefore it is better to write "in increasing concentrations of CuSO4" in the Table description
Fig. S4-S8 Authors should decide whether to use Gram-negative or gram-negative or gram negative, the same applies to Gram-positive
Fig. S4 / S11 "ef1" should be added to Marinomonas and Brevundimonas
Author Response
We thank the Reviewer for the precious suggestions.
Comment: I also have two questions for the Authors:
Out of curiosity, I wonder if the authors have tested these bacterial strains produced CuONPs for cytotoxicity? Any data showing low toxicity especially on human cells would be nice to include and strengthen their potential as future antibacterial agents. The authors only gave in their conclusions link to the publication where L50 values for C. reinhardtii, E. coli, D. magna and embryonic zebrafish were estimated.
Response: We are working on it and this analysis requires a precise experimental design and a series of tests, for this reason it will be the topic of a separate paper.
As the authors conclude that antimicrobial activity is likely due to ROS induction, a simple test with DHE or H2DCFDA would be a good complement to the microbiological analysis. Authors should think about this in the future.
Response: we thank the reviewer for the nice suggestion. We would prefer to maintain the focus of this work on the description of nanoparticle biosynthesis and antimicrobial activity detection. We will perform analysis on the ROS activity in a next paper.
Suggested changes
- comparing Table 1 with Table 2, it seems to me that the MTCs for Rhodococcus ef1 should be 4.5 mM and for Brevundimonas ef1 - 5 mM
Response: we changed as suggested
- all bacterial species should be in italics (line 97, 103, 104, 116, 201-204, 304-306, Fig. 4)
Response: in the revised version we corrected all names
- Fig. 1, the X and Y axis signatures are not very legible and should be larger and on the graph for BC-ef1 s “nm" is missing
Response: we corrected as suggested
- line 219, I would remove "by serial dilution"
Response: we corrected as suggested
- authors should decide whether to use μg/ml or μg/mL (line 224 and 228)
Response: we corrected all mL into ml
- Table 5, the description is illogical, should be “MIC and MBC values (μg/ml) of the CuONPs synthesized by MM-ef1, RH-ef1, PM-228 ef1, BM-ef1 and BC-ef1 against pathogenic bacteria“
- Methods - MIC evaluation, it is more logical that positive control is medium with inoculum and negative control only medium. It should be changed
- Appendix A, Table “IR assignments signatures” - should not be in italics
- Supplementary Figures:
Fig. S1 Not for all strains the CuSO4 concentration was 0.5-4 mM. The concentrations are given in the diagram, therefore it is better to write "in increasing concentrations of CuSO4" in the Table description
Fig. S4-S8 Authors should decide whether to use Gram-negative or gram-negative or gram negative, the same applies to Gram-positive
Fig. S4 / S11 "ef1" should be added to Marinomonas and Brevundimonas
Response: we followed all suggestions
Reviewer 4 Report
The authors of the manuscript conclude that the ability of five Antarctic bacterial strains to synthesize CuONP may be a defense mechanism against this heavy metal. Did the authors check whether the same species of bacteria that they studied, but not originating from Antarctica, do not have this mechanism and are not resistant to 5 mM CuSO4? Would copper nanoparticles obtained by using another strain of Bacillus or Pseudomonas (e.g. isolated from soil in Europe) have a similar ability to synthesize CuONP?Due to the fact that the authors did not explain the mechanism of action of copper nanoparticles on pathogenic bacteria, they should at least prove whether the Antarctic bacteria used for the synthesis of CuONP are unique. The very method of green synthesis of copper nanoparticles is already known. Authors must demonstrate the uniqueness of their method. The conclusions should not cite the literature, because it is a summary of the results obtained by the authors.Why was 1 mM CuSO4 x 5H2O solution used for the synthesis of nanoparticles?Authors wrote that: “It was observed that the bactericidal property of CuONPs is due to the generation of large number of small molecules containing reactive oxygen species (ROS) by the nanoparticles attached to the bacterial cells” (lines 230 – 232). I have a question - who observed and where?The species of bacteria should be written in italics (e.g. lines 97, 201 – 204, 209 – 212 etc.)There is no description of X axis and Y axis in Fig. S1.Citation No 21 concerns silver nanoparticles and not copper nanoparticles.
Author Response
We thank the Reviewer for the precious suggestions.
Comment: The authors of the manuscript conclude that the ability of five Antarctic bacterial strains to synthesize CuONP may be a defense mechanism against this heavy metal. Did the authors check whether the same species of bacteria that they studied, but not originating from Antarctica, do not have this mechanism and are not resistant to 5 mM CuSO4? Would copper nanoparticles obtained by using another strain of Bacillus or Pseudomonas (e.g. isolated from soil in Europe) have a similar ability to synthesize CuONP?
Response: non Antarctic bacteria have been reported to be able to synthesize CuNPs or CuONPs, such as Serratia (10.1166/jnn.2008.095 ), Pseudomonas fluorescens (Shantkriti and Rani, International Journal of Current Microbiology and Applied Sciences 3(9):374-383) or Bacillus cereus (http://dx.doi.org/10.1016/j.procbio.2016.08.008). This latter is able to tolerate >10 mM of copper and it is able to synthesize CuNP.
What we wanted to conclude in this paper is that even in Antarctica where the levels of metal contamination is low, bacteria became tolerant and can acquire mechanism for resistance such as the synthesis of nanoparticles. We do not intend to make a comparison with not cold adapted species, we only try to give an explanation why also Antarctic bacteria are metal resistant and are able to produce metal nanoparticles. Furthermore, these Antarctic bacteria grow and are able to synthesize NPs at lower temperatures than those from non-Antarctic habitat, therefore, is difficult to make a comparison in their ability to synthesize NPs.
Comment: Due to the fact that the authors did not explain the mechanism of action of copper nanoparticles on pathogenic bacteria, they should at least prove whether the Antarctic bacteria used for the synthesis of CuONP are unique. The very method of green synthesis of copper nanoparticles is already known. Authors must demonstrate the uniqueness of their method.
Response: These bacteria are unique because are able to synthesize CuONPs very quickly at temperatures lower than those used by other microorganisms and starting from a low concentration of salt (1 mM). Anyway, the focus of the paper is not to provide a new method/application of nanoparticle production but to describe a new case of nanoparticle biosynthesis from Antarctic bacteria
Comment: The conclusions should not cite the literature, because it is a summary of the results obtained by the authors. Why was 1 mM CuSO4 x 5H2O solution used for the synthesis of nanoparticles?Authors wrote that: “It was observed that the bactericidal property of CuONPs is due to the generation of large number of small molecules containing reactive oxygen species (ROS) by the nanoparticles attached to the bacterial cells” (lines 230 – 232). I have a question - who observed and where?
Response: The conclusion has been modified as suggested. The deleted sentences are added in the introduction.
We used 1 mM of CuSO4 x 5H2O because the concentration of salt may cause differences in the size of NPs. Since we obtained good results with 1 mM, we kept this concentration.
In the paper “Applerot, G., Lellouche, J., Lipovsky, A., Nitzan, Y., Lubart, R., Gedanken, A. and Banin, E. , Understanding the Antibacterial Mechanism of CuO Nanoparticles: Revealing the Route of Induced Oxidative Stress. Small, 2012. 8, 3326-3337” (36 in the paper) is reported this observation.
Comment: The species of bacteria should be written in italics (e.g. lines 97, 201 – 204, 209 – 212 etc.) There is no description of X axis and Y axis in Fig. S1. Citation No 21 concerns silver nanoparticles and not copper nanoparticles.
Response: in the revised version of the paper, we corrected these mistakes.
Round 2
Reviewer 1 Report
The article looks much better now and the authors have accepted the suggestions and improved the review article. I have no further comments.
Reviewer 4 Report
After corrections, the manuscript can be accepted for publication in the journal.